# Endogenous Bornavirus-like Elements in Bats: Evolutionary Insights from the Conserved Riboviral L-Gene in Microbats and Its Antisense Transcription in *Myotis daubentonii*

**DOI:** 10.3390/v16081210

**Published:** 2024-07-27

**Authors:** Muriel Ritsch, Tom Eulenfeld, Kevin Lamkiewicz, Andreas Schoen, Friedemann Weber, Martin Hölzer, Manja Marz

**Affiliations:** 1RNA Bioinformatics and High-Throughput Analysis, Friedrich Schiller University Jena, 07743 Jena, Germany; 2European Virus Bioinformatics Center, 07743 Jena, Germany; 3Cluster of Excellence Balance of the Microverse, Friedrich Schiller University Jena, 07743 Jena, Germany; 4Institute for Virology, FB10-Veterinary Medicine, Justus Liebig University, 35392 Gießen, Germany; 5Genome Competence Center (MF1), Robert Koch Institute, 13353 Berlin, Germany; 6German Centre for Integrative Biodiversity Research (iDiv) Halle-Jena-Leipzig, 04103 Leipzig, Germany; 7Fritz Lipmann Institute-Leibniz Institute on Aging, 07745 Jena, Germany

**Keywords:** bat, viruses, endogenous viral elements, EVEs, endogenous bornavirus-like L element, EBLL, *Cultervirus*, *Myotis daubentonii*, antisense transcript

## Abstract

Bats are ecologically diverse vertebrates characterized by their ability to host a wide range of viruses without apparent illness and the presence of numerous endogenous viral elements (EVEs). EVEs are well preserved, expressed, and may affect host biology and immunity, but their role in bat immune system evolution remains unclear. Among EVEs, endogenous bornavirus-like elements (EBLs) are bornavirus sequences integrated into animal genomes. Here, we identified a novel EBL in the microbat *Myotis daubentonii*, EBLL-Cultervirus.10-MyoDau (short name is CV.10-MyoDau) that shows protein-level conservation with the L-protein of a *Cultervirus* (Wuhan sharpbelly bornavirus). Surprisingly, we discovered a transcript on the antisense strand comprising three exons, which we named AMCR-MyoDau. The active transcription in *Myotis daubentonii* tissues of AMCR-MyoDau, confirmed by RNA-Seq analysis and RT-PCR, highlights its potential role during viral infections. Using comparative genomics comprising 63 bat genomes, we demonstrate nucleotide-level conservation of CV.10-MyoDau and AMCR-MyoDau across various bat species and its detection in 22 *Yangochiropera* and 12 *Yinpterochiroptera* species. To the best of our knowledge, this marks the first occurrence of a conserved EVE shared among diverse bat species, which is accompanied by a conserved antisense transcript. This highlights the need for future research to explore the role of EVEs in shaping the evolution of bat immunity.

## 1. Introduction

Within all vertebrates, bats (*Chiroptera*) are the most abundant and ecologically diverse animals [1]. Except for the polar regions, bats are globally distributed [2] and their origin has been dated in the Cretaceous period [3]. In addition to having the ability to fly [3], to learn to produce vocalizations by hearing [4], and an exceptional longevity [5,6,7], bats are also known for their unique natural resistance to many pathogenic viruses [1,8,9,10]. For example, MERS and Ebola RNA viruses cause fatal infections in humans; however, bats appear to be asymptomatic and survive the infection [11,12,13,14]. A new type of coronavirus, SARS-CoV-2, led to a worldwide pandemic starting in 2019 and caused more than 6.9 million deaths, according to the WHO (www.who.int/emergencies/diseases/novel-coronavirus-2019 (accessed on 29 April 2024)) and most probably also originated in bats [15,16]. Despite the unique biological characteristics of these flying mammals, bats are one of the least studied taxa of all mammalian [2,17,18], and the reasons for their function as reservoirs for various viruses remain unclear [19,20,21]. However, bats’ uniquely developed immune systems may provide the solution to better understanding and fighting various pathogens and preventing future pandemics [13,22,23].

The immune systems of various bat species have been studied in the past (see [9] for an overview), with most focusing on differently expressed (protein-coding) genes [13,17,24,25,26]. Genes related to or associated with the immune system are typically differently expressed during viral infections on extreme levels [27]. In bat cells, these expression changes often appear not as drastically as in human cells, as previously shown in a comparison of a human and bat cell line infected with Ebola and Marburg virus [13]. Although investigations into the bat transcriptome during viral infections provide first insights into unique features, it is still elusive how the immune system of bats functions as a whole to control viral infections and how important key players act together. In addition, there is still little knowledge about the non-protein-coding transcriptome of bats, even though noncoding RNAs (ncRNAs) also play important roles during infections in bats [5,28,29].

One part of the ncRNA transcriptome consists of transcribed endogenous viral elements (EVEs), which have recently become more and more focused in the context of infections [30,31]. When a virus infects a cell, a part of the viral genome can be integrated into the host genome. In germline cells, integrated viral fragments can become EVEs; this process is called endogenization [32]. Once integrated into the host genome, EVEs can increase their allele frequency via vertical transmission until their potential fixation [32]. The first described EVEs were derived from endogenous retroviral elements (ERVs) [33,34]. For retroviruses, the integration of the retroviral RNA genome into the host DNA is a major step in the replication cycle [35]. Although less common, EVEs originating from nonretroviruses have been identified across many eukaryotic organisms, including unicellular eukaryotes, plants, vertebrates, and arthropods [33,36,37,38,39,40,41,42,43,44]. For example, EVEs derived from bornaviruses (nonsegmented, negative strand RNA viruses) were detected in eukaryotic genomes [45]. The family *Bornaviridae* comprises three genera: *Carbovirus*, *Cultervirus*, and *Orthobornavirus*, with EVEs present in all genera [46]. Different types of endogenous bornavirus-like (EBL) elements exist depending on which part of the viral mRNA has been integrated. To date, the elements of EBLN (nucleoprotein), EBLM (matrix protein), EBLG (envelope glycoprotein), EBLL (RNA-dependent RNA polymerase), EBLP (phosphoprotein), and EBLX (accessory protein) are known [47,48,49]. Despite being RNA viruses, bornaviruses likely have their mRNA reverse transcribed and integrated into the host genome via the action of the long interspersed nuclear element 1 (LINE-1) retrotransposon [33,45].

Among other mammals, EVEs with intact open reading frames (ORFs) have already been described in bat species. For instance, in *Eptesicus fuscus*, an EBLL element (efEBLL-1) was identified with an intact and complete ORF spanning 1718 codons [47], thus marking the initial observation of an EVE capable of encoding a functional RNA-dependent RNA polymerase. The function of this protein remains unclear, as is the case with most described EVEs. A notable exception is the VP35-like ORF derived from ancient filoviruses, where a hypothetical regulatory function has been proposed. Interestingly, this VP35-like protein exhibits an antagonist function to its exogenous homolog [30], thus potentially serving as a regulator of innate immune signaling. Various investigations have been conducted regarding EVEs in bats and their function, yet more than speculation or vague assertions are seldom made. It is assumed that some of these EVEs in bats contribute to bat-specific immune mechanisms that may confer a virus-tolerant phenotype [21,50]. In other organisms as well, these functions are typically inferred rather than definitively established [51]. An exception is the well-described syncytins, which are found, for example, in humans and mice, thus originating from the envelope gene of an expressed endogenous retrovirus [52,53,54,55,56,57].

In this work, we describe a novel EBLL in *Myotis daubentonii* (EBLL-Cultervirus.10-MyoDau) and its homologs in 18 bat species, thus following the EVE nomenclature proposed by Kawasaki et al. [48]. For better readability, we refer to EBLL-Cultervirus.10-MyoDau as CV.10-MyoDau in the following, and we omit the host component (-MyoDau) if we are talking about the locus in general and not about a copy within a species.

We demonstrate conservation at the nucleotide level and similarities at the protein level. Unexpectedly, an antisense transcript composed of three exons, which we named Antisense Myotis Complementary RNA (AMCR-MyoDau), has been confirmed through RNA-Seq analysis and RT-PCR in a *Myotis daubentonii* cell line. This AMCR-MyoDau is partly conserved in 22 *Yangochiropera* and 12 *Yinpterochiroptera* bat genomes.

Our results pave the way for further studies on transcribed EVEs to elucidate their potential function and role in virus immunity—even beyond bat species.

## 2. Materials and Methods

### 2.1. Bat Genome References and Gene Annotation Construction

We scanned genomes of 63 bat species for potential EVEs, thus comprising 42 of the suborder *Yangochiroptera* and 21 of *Yinpterochiroptera*, which were obtained from the NCBI database [58] and are of of varying assembly qualities (Table 1). Gene annotations were only available for reference sequences (RefSeqs, marked with an asterisk in Table 1). We performed liftover annotations for the other genomes lacking annotations using Liftoff (v1.6.3) [59]. We used the reference gene annotation from *Myotis daubentonii* (see Table 1 for species abbreviations) to annotate genes for all microbats lacking an annotation. *Myotis daubentonii* is one of the more recently published (20 August 2023) reference genomes within microbats, which is characterized by a small number of contigs and even annotated chromosomes. We used the reference annotation from *Rousettus aegyptiacus* for megabats, thus benefiting from its assembly composed of only 29 contigs, which is the most developed assembly available for megabats. The phylogenetic tree was adopted from the study by Agnarsson et al. [60]; see Appendix A.

### 2.2. Processing of RNA-Seq Data, Refinement of CV.10-MyoDau Annotation, and Differential Gene Expression

We used tblastn v2.15.0+ (E-value < $10^{-6}$) to search for new EVEs in the genome of *Myotis daubentonii* using the amino acid sequence of the L protein of a RefSeq Wuhan sharpbelly bornavirus of the genus *Cultervirus* (YP_010085030.1) as query (see https://www.doi.org/10.17605/OSF.IO/89EF2 for tblastn results). One of the hits was defined as EBLL-Cultervirus.10-MyoDau (NC_081844.1: 39,922,667–39,923,434; short name: CV.10.MyoDau), thus following the EVE nomenclature defined by Kawasaki et al. [48]. For transcriptome analyses, we used the genome assembly of *Myotis daubentonii* (mMyoDau2.1, Table 1) with the corresponding annotation (GCF_963259705.1) as a reference and utilized reads from our previously published RNA-Seq dataset derived from a virus- and an interferon alpha-induced *M. daubentonii* cell line (https://www.ncbi.nlm.nih.gov/bioproject/?term=GSE121301 (accessed on 1 July 2024)) [23]. From this study, we downloaded the quality-trimmed and rRNA-cleaned reads in FASTQ format for further downstream analyses (https://doi.org/10.17605/OSF.IO/X9KAD). In total, we obtained 18 short-read, single-end, and strand-specific Illumina samples comprising two postinfection time points (6 h, 24 h) and three conditions (mock, interferon alpha induction, Rift Valley fever virus Clone13 infection) in three biological replicates.

First, to verify our initial homology-based identification of CV.10-MyoDau and to more precisely define exon boundaries for this novel EVE, we mapped the RNA-Seq data to the *Myotis daubentonii* reference genome with HISAT2 v2.1.0 using default parameters [61,62] and investigated the mapping results using IGV [63]. For a programmatic approach, we used SAMtools to extract the depth at each nucleotide position from the mapped reads for the 18 samples. We determined the exon boundaries based on the sum of reads exceeding 10 across all 18 samples and refined them with split reads. The definition of the three resulting exons in Gene Transfer Format (GTF) format and the associated sequences of the exons belonging to the AMCR-MyoDau are available in the https://www.doi.org/10.17605/OSF.IO/89EF2.

Next, we ran RNAflow v1.4.6 [64] for mapping, read count normalization, and conducted differential gene expression analysis via DESeq2 [65] with parameters to skip the rRNA depletion step and employed counting reads in reverse strand specificity to match the RNA-Seq library design. We manually extended the *Myotis daubentonii* annotation GTF file by adding gene, transcript, and exon features for the three exons of AMCR-MyoDau. We obtained the adjusted *p* values, log2-fold change values, and normalized expression counts for the three exons and all pairwise comparisons.

### 2.3. In Vitro Validation of EVE Candidate

Cells were either mock treated, infected with Rift Valley fever virus (RVFV) Clone13 (MOI 5), or treated with 1000 U/mL pan species type I interferon (IFN). At the indicated time point, total cellular RNA was isolated using RNeasy Mini Kit (Qiagen, Cat No./ID: 74106) according to the manufacturer’s instructions. A total of 100 ng isolated RNA was used for random primed cDNA synthesis using PrimeScript High Fidelity RT-PCR Kit (Takara, R022B). PCR was conducted using a primer set designed using the online software Primer3Plus [66] to detect the AMCR-MyoDau 90 nucleotide sequence of exon 3 (fwd primer—CTCCCTTGAGGAGTGTGACC—and rev primer—GGCTGTCAGCAACAGTTTCA), KOD polymerase (Calbiochem, 71,086.3), and 2 µL of the respective cDNAs as templates. The annealing temperature was set to 57.1 °C for 35 cycles, and the resulting DNA products were separated on a 2% agarose (Serva, 11,404.07) in TAE (40 mM Tris, 20 mM Glacial acetic acid, 1 mM EDTA) buffer gel for 30 min at 100 V. The amplicons were then visualized using incubation in 0.25 µg/mL ethidium bromide in H_2_O for 20 min, and pictures were taken using the BioRad ChemiDoc MP imaging system.

### 2.4. Synteny Analysis to Confirm the Conservation of AMCR-MyoDau and CV.10-MyoDau

As a template, we extracted the three nucleotide sequences of the transcriptome-refined exons of AMCR-MyoDau and the authentic CV.10-MyoDau region from the *Myotis daubentonii* genome and searched them in all other selected bat genomes (Table 1) using blastn v2.15.0+ (E-value < $10^{-6}$) [67]. Since CV.10-MyoDau is part of exon 3 of AMCR-MyoDau, the synteny analysis of this EVE is similar to the synteny of AMCR-MyoDau. We conducted a synteny analysis for all blastn hits using either the annotations from the reference genomes or the built liftovers from the bat genomes without appropriate references. For this purpose, we recorded all genes within 1 million nucleotides in the downstream and upstream directions; for annotations of all surrounding genes, see https://www.doi.org/10.17605/OSF.IO/89EF2. The distances between the genes and the gene lengths were logarithmically transformed for visualization. A table containing all nucleotide blastn hits from the three exons of AMCR-MyoDau and CV.10-MyoDau, and their associated synteny is available in the https://www.doi.org/10.17605/OSF.IO/89EF2 as a feature file in GFF format and a human- and machine-readable plain text file.

Additionally, we provide truncated versions of these files containing only the blastn hits associated with a curated subset of syntenies.

### 2.5. Identification and Further Investigation of the Previously Described EBLL-IG

We refer to the homologous family of identified genes derived from AMCR-MyDau as AMCR. Due to numerous hits beyond AMCR’s synteny, we created a coverage plot of exon 3 for each synteny unit of all blastn hits to explore additional EVEs; see Appendix A. The coverage plot was created for exon 3 on the nucleotide level, as only exon 3 exclusively overlapped with CV.10-MyoDau. This plot unveiled another conserved EVE (sequence NC_081852.1 indices 16,169,651–16,170,474), as previously described in the literature as EBLL intergenus (EBLL-IG) [68]. With the resulting EBLL-IG blastn hit from exon 3 in *Myotis daubentonii*, additional searches were conducted using blastn, which are similar to exons 1–3 of AMCR-MyoDau, followed by a synteny analysis of the additional hits for the FASTA template and the blastn results; see https://www.doi.org/10.17605/OSF.IO/89EF2. Given the similarity between the templates of exon 3 of AMCR-MyoDau and EBLL-IG, it is possible that blastn hits for both templates could be found in the same source region. In this scenario, we displayed only the longer hit in the synteny plot.

### 2.6. Multiple Sequence Alignments of the Three Exons of AMCR in Various Bat Species

For the three AMCR exons identified in other bat species through synteny confirmation, we generated multiple nucleotide sequence alignments using MAFFT v7.520 employing pairwise alignments with Smith–Waterman algorithm (L-INS-i strategy, parameters: --auto, --localpair. --maxiterate 1000) [69]. For exon 3, a more advanced alignment procedure was required. We took the nucleotide sequences from all individual synteny-confirmed exon 3 blastn hits and aligned them against the exon 3 template from *Myotis daubentonii* using the Smith–Waterman algorithm implemented in MAFFT (L-INS-fragment strategy, parameters: --localpair, --maxiterate 1000, --addfragments). This study defined sequences in the alignment belonging to the same contig within a distance of fewer than 2000 nucleotides as *related*. To preserve the alignment, their corresponding sequences were concatenated using three ambiguous nucleotide bases (`NNN’). If neighboring hits in the respective bat overlapped in the query, specific nucleotides were eliminated, and the sequences were concatenated with the NNN block. The alignments are available as FASTA files with gaps indicated by a “ -” character in the https://www.doi.org/10.17605/OSF.IO/89EF2. The concatenated alignment of the three AMCR exons was visualized using CIAlign (parameter: --visualise) [70].

## 3. Results

### 3.1. A Novel EVE and Its Antisense Transcription in Myotis daubentonii: EBLL-Cultervirus.10-MyoDau and AMCR-MyoDau

After an extensive literature screening, to the best of our knowledge, we present a previously unreported EBLL locus in *Myotis daubentonii*, which we call EBLL-Cultervirus.10-MyoDau (short name is CV.10-MyoDau), originating from the L gene of a *Cultervirus* (YP_010085030.1) [45,47,48,68,71,72] (Figure 1). The CV.10-MyoDau is situated on the plus strand on chromosome 5 of *Myotis daubentonii* (NC_081844.1: 39,922,667–39,923,434). Three directly overlapping sequences with the L protein as query were identified using tblastn, with an E value of $1.57\times 10^{-34}$. These sequences resulted in a protein alignment that is 268 amino acids long between the L protein and CV.10-MyoDau, thus requiring two frameshifts. The longest of the three sequences comprises 176 amino acids, but it contains several stop codons, which suggests compromised functionality; see Figure 1.

To investigate a possible transcription of this novel EBLL locus in *Myotis daubentonii*, we reanalyzed the RNA-Seq data from a previous study [23]. Here, we had previously examined the transcriptional landscape of a virus- and an interferon alpha-induced *Myotis daubentonii* cell line at two time points postinfection (6 h and 24 h). Thus, these data set were well suited for rescreening for potential EBLL-derived transcripts, especially since the sequencing was performed in a strand-specific manner, thus allowing us to detect whether transcripts originated from the plus or minus strand. Suprisingly, we identified a transcript on the minus strand consisting of three exons; see Figure 1. This has been identified as an antisense EVE, which is henceforward referred to as AMCR-MyoDau. Collectively, these three exons span a genomic length of 1903 nucleotides, which are distributed across a 6 kb genome region. Only a specific segment of exon 3 overlaps with CV.10-MyoDau; logically, only this part shares similarities with the L protein of the Wuhan sharpbelly bornavirus. This implies that there is only partial overlap between AMCR-MyoDau and CV.10-MyoDau; see Figure 1.

### 3.2. AMCR-MyoDau Shows Weak but Constant Expression in a Myotis daubentonii Cell Line under Mock, Interferon-Induced, and Virus-Infected Conditions

Next, we used the RNA-Seq data to refine the exon boundaries of CV.10-MyoDau and to investigate the expression levels. AMCR-MyoDau showed a shallow but constant expression, with TPM values between 4.3 and 9.5 in all the investigated RNA-Seq samples and conditions. This was confirmed using strand-specific RNA-Seq analysis (Figure 2). Random-primed RT-PCR confirmed the RNA expression through an independent method (Figure 3). We generated a coverage plot for all samples, where the sum of all mapped reads was scaled by 10, which also formed the basis of defining the exon boundaries for AMCR-MyoDau in *Myotis daubentonii*; see Figure 2. After further refinement employing spliced reads, we obtained the exon features at positions 39,922,089–39,923,568 (exon 3), 39,924,253–39,924,371 (exon 2), and 39,929,747–39,930,053 (exon 1). Please note the inverse order of the exon numbering due to the identification of AMCR-MyoDau on the minus strand (5′ to 3′); see Figure 1. Additionally, we found AMCR-MyoDau to be significantly (adjusted *p* value < 0.05) differentially expressed in four pairwise comparisons; see https://www.doi.org/10.17605/OSF.IO/89EF2. The comparison *24 h Mock* vs. *24 h Clone13* showed the highest log2-fold change of −0.93, thus indicating the downregulation of AMCR-MyoDau. We hypothesize that the decline observed after 24 h for Clone13 is most likely due to cap snatching, which involves the cleaving off of host RNA 5′ caps by the viral polymerase, thus leading to host cell shutoff [73].

We investigated further whether AMCR-MyoDau is transcribed using RT-PCR with RNA samples from our *Myotis daubentonii* cell line [23]. As the RNA-Seq results indicate, the expected bands were detected under all the available conditions, including mock, RVFV Clone13 infection, and IFN-Alpha treatment; see Figure 3. Therefore, AMCR-MyoDau is expressed as RNA in *Myotis daubentonii* and is thus an antisense transcript of the EVE.

### 3.3. Synteny Patterns of AMCR and CV.10-MyoDau across Various Bat Species

For a synteny analysis, we first screened the nucleotide sequences of the three exons of AMCR-MyoDau across all listed bats using blastn to find the relevant positions in the bat genomes; see Table 1. As CV.10-MyoDau was found to be part of exon 3, a direct investigation of the synteny of this newly described EVE was also conducted. In total, 311 individual sequences were identified on nucleotide similarity across 42 distinct bat species. After concatenating the *related* hits nearby, 277 sequences remained. For all these locations, a synteny analysis was performed, and hits from various bats that exhibited the same synteny pattern were grouped together.

In total, we observed 29 different syntenic patterns in the nucleotid context of AMCR-MyoDau and CV.10-MyoDau. All these regions could be further evaluated as potential EVEs. But, there were just two syntenic patterns covering the CV.10-MyoDau region; see Figure 4 and Appendix A. Appendix A presents all syntenic blocks unrelated to CV.10-MyoDau that are not shown in Figure 4.

The first syntenic pattern we identified has been named AMCR after the transcript. We could identify a consistent syntenic pattern (*GALNT7*, *HMGB2*, AMCR, *SAP30*, and *SCRG1*) for at least 34 bats. This indicates that the AMCR-MyoDau gene is at least partly conserved in at least 34 other bat species, although we cannot confirm its potential transcription activity. We propose naming these genes as AMCR followed by the specific bat species (e.g., AMCR-EptFus for *Eptesicus fuscus*). In total, 72 nucleotide sequences (after removing recurring hits in the same source region) were identified. A definitive set comprising 24 sequences for exon 1, 17 for exon 2, and 31 for exon 3, recognized as true positive sequences for AMCR, were selected for subsequent processing. For only 18 bat species within this syntenic pattern, sequence similarities to CV.10-MyoDau were identified. This means that these 18 EVEs are orthologs to CV.10-MyoDau and should be named according to the adjusted ERV nomenclature (e.g., CV.10-EptFus for *Eptesicus fuscus*) [74]. Out of these 18, 8 of them have already been detected using automated pipelines in *Eptesicus fuscus*, *Antrozous pallidus*, *Pipistrellus kuhlii*, *Pipistrellus pipistrellus*, *Murina aurata feae*, *Myotis lucifugu*, *Myotis brandtii*, and *Myotis myotis* [48,72]. These pipelines utilized outdated accessions, thus making the direct transfer a challenge. Another challenge lies in the high sequence similarities among paralogous EBLLs. To ensure that the EVEs found by the pipeline are indeed the same, we examined the genomic context.

We have identified nine additional novel orthologs, alongside our novel CV.10-MyoDau, specifically in the following bat species: *Aeorestes cinereus*, *Corynorhinus townsendii*, *Plecotus auritus*, *Ia io*, *Eptesicus nilssonii*, *Pipistrellus pygmaeus*, *Myotis vivesi*, *Myotis yumanensis*, and *Myotis ricketti*. The specific positions in the respective sequences of the genomes can be found in Table 2 and in the https://www.doi.org/10.17605/OSF.IO/89EF2.

The second syntenic pattern, sharing sequence similarities to CV.10, is the already described EBLL-IG [68]. EBLL-IG sequences were identified as blast hits using the L protein of Wuhan sharpbelly bornavirus, which is a paralog of CV.10-MyoDau. While EBLL-IG has previously exclusively been characterized in *Myotis davidii* and *Eptesicus fuscus* to date, our findings unveil its occurrence across 13 bat species in the *Vespertilionidae* family. Additionally, we have also evaluated conservation at the synteny level. An alignment is available in the https://www.doi.org/10.17605/OSF.IO/89EF2. Based on the sequence similarity, EBLL-IG seems to be a duplication of CV.10-MyoDau for all *Vespertilionidae* and was deleted in a later evolutionary event for a subset of *Myotis* species (*Myotis lucifugus*, *Myotis brandtii*, *Myotis vivesi*, and *Myotis yumanensis*). However, we can only hypothesize about the duplication event and cannot exclude the possibility that independent endogenization events of similar viruses generated the EBL elements.

In addition to the two EVEs presented here (CV.10-MyoDau and EBLL-IG), there are also numerous other EBLLs [21,45,47,48,68,72]. It needs to be determined which EBLL was integrated first or if there were multiple independent integration events.

In the depicted synteny plot of CV.10 (Figure 4), homologs of exon 3 have only been found for *Vespertilionidae* in the *SCRG1* gene (in the purple box). However, in the coverage plot in Appendix A (purple line), we observed that these sequences do not share any commonality with the L protein. Therefore, they do not represent the EVE itself but a potential further development of the functional unit derived from CV.10. This syntenic block is only interesting because of its proximity to CV.10.

On the other hand, nine homologous L protein samples could be identified. However, they could not be assigned to any synteny, as illustrated in the red-marked box in Figure 4 and as a red line in Appendix A. These nine 600 nt long potential homologous genes with an E value smaller than $6\times 10^{-26}$ belong exclusively to *Nycticeius humeralis*, *Lasiurus borealis*, *Murina aurata feae*, and *Myotis vivesi*, with all of them being members of the *Vespertilionidae* family. Controversially, these four bat species are phylogenetically not closely related, but their genomes are highly fragmented (see the number of contigs and the N50 value in Table 1). Therefore, the overall relation to the L protein remains questionable.

### 3.4. Comparative Genome Analysis Shows AMCR Sequence Similarities in 34 Bat Species

Due to synteny similarity, a total of 72 sequences from 34 species were identified as true positive sequences for AMCR (24 sequences for exon 1, 17 for exon 2, and 31 for exon 3). With these sequences, separate nucleotide alignments were built for each exon of AMCR; see the https://www.doi.org/10.17605/OSF.IO/89EF2. We used CIAlign to visualize the nucleotide alignments of all three exons. Figure 5 illustrates the distribution of exons among different bat species, thus providing detailed information about the specific regions and conservation within each exon. Generally, a high conservation can be observed in the aligned nucleotide sequences: the identity of nongap nucleotides with the consensus sequence is 91% in exon 1, 97% in exon 2, 96% in exon 3, and 95% for the combined exons. Furthermore, it is evident that the region of exon 3 of AMCR corresponding to the CV.10 EVE is highly conserved in *Vespertilionidae*.

### 3.5. Partial Occurrence of AMCR across Multiple Bat Families and Evolutionary Conservation of CV.10 in Vespertilionidae

The exon structure of AMCR-MyoDau and conservation to the L protein seem to be preserved in the family of *Vespertilionidae*, as revealed by synteny and sequence similarity analyses (Figure 4 and Figure 5). However, we could not detect the entire exon 3 of AMCR-MyoDau for a few other homologs within *Yangochiroptera*, and no matches were found for *Yinpterochiroptera*. This suggests that the L protein part integrated only after the divergence of the *Vespertilionidae* approximately 36–47 mio years ago, but it would have been before further branching inside the *Vespertilionidae* family 16–20 mio years ago [60,75]. CV.10 was not found in four *Vespertilionidae* species (*Myotis davidii*, *Lasiurus borealis*, *Nycticeius humeralis*, and *Myotis vivesi*). In *Myotis davidii*, exon 1, exon 2, and parts of exon 3 of AMCR-MyoDav were identified. However, *Myotis davidii*’s assembly quality is poor, with over 100 thousand contigs and an N50 value of 3.5 Mb. In the other three bat species (*Lasiurus borealis*, *Nycticeius humeralis*, and *Myotis vivesi*), sequences were found with CV.10, but these were not confirmed by synteny, thus likely due to relatively poor assembly quality with very high contig numbers and small N50 values; see Table 1.

In addition, the presence of exon 1 in nearly all bat species implies that this functional segment likely predated the divergence of *Yangochiroptera* and *Yinpterochiroptera*.

## 4. Discussion

Bats (*Chiroptera*) are renowned for their ecological diversity and unique traits—including resistance to pathogenic viruses—and they remain understudied despite their potential as viral disease reservoirs. The recent emergence of SARS-CoV-2 highlights the urgent need to comprehend bat biology and immunity.

Our discovery of the novel endogenous bornavirus-like element (CV.10-MyoDau) in *Myotis daubentonii* bats, with conserved nucleotide and protein sequences, provides insight into the intricate relationship between bats and viruses. Surprisingly, we found an antisense transcript, named AMCR-MyoDau, comprising three exons, as confirmed by RNA-Seq and RT-PCR analysis in *Myotis daubentonii* tissues. In the bat *Myotis daubentonii*, the causality between the transcription originating from the minus strand and the conservation of the L protein situated on the plus strand is uncertain. The functions of CV.10-MyoDau or AMCR-MyoDau remain speculative at this time. This represents the first instance of a conserved EVE and accompanying antisense transcript among diverse bat species, thus underlining the need for further investigation into the role of EVEs in bat immunity evolution. The three exon sequences of AMCR-MyoDau, as well as the region of CV.10, are preserved in *Vespertilionidae*. In total, there are 17 orthologs of CV.10-MyoDau, which we confirmed through synteny analysis, with 9 of these orthologs being novel.

Future studies should prioritize unraveling the complexities of bat immune responses, thus considering the potential role of transcribed EVEs in influencing bat immunity against viral infections. The observed virus-tolerant phenotype in bats may be associated with specific adaptations, thereby possibly manifested by the diversity and abundance of various EVEs described in these flying mammals.

However, one primary challenge in the still-young field of EVE research is the absence of a standardized database and widely used nomenclature for their cataloging. Currently, there is no centralized repository documenting previously identified EVEs, thus complicating the process of verifying whether a newly discovered EVE is truly novel. As researchers who frequently search for novel EVEs, we experience this challenge firsthand, thus underscoring the urgent need for a collaborative initiative to establish such a repository.

## Figures and Tables

**Figure 1 viruses-16-01210-f001:**
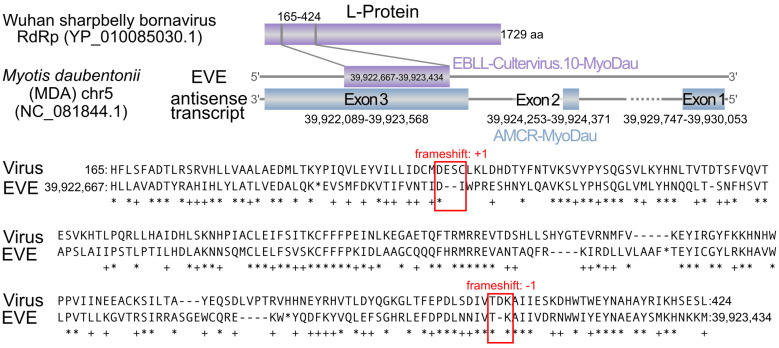
**Top:** A schematic illustration is provided for the newly described EBLL-Cultervirus.10-MyoDau (CV.10-MyoDau) alongside the L protein encoded by the RNA-dependent RNA polymerase (RdRP) of Wuhan sharpbelly bornavirus. CV.10-MyoDau is located on the plus strand. There is also an antisense transcript comprising three exons on the minus strand, which is named AMCR-MyoDau. **Bottom:** Amino acid sequence alignments of the L protein of Wuhan sharpbelly bornavirus and CV.10-MyoDau: The red boxes indicate frameshifts. In the amino acid sequences, “-” represents gaps in the alignment, and “*” denotes stop codons. Below the two amino acid sequences, “*” indicates identical amino acids, while “+” signifies similar amino acids according to the BLOSUM62 matrix.

**Figure 2 viruses-16-01210-f002:**
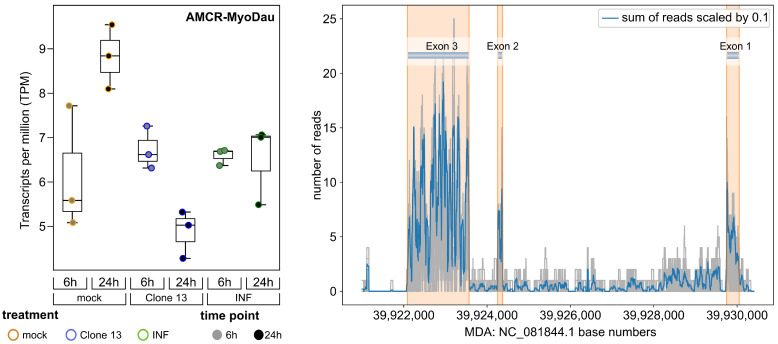
**Left:** Expression box plot of AMCR-MyoDau. The plot shows the TPM values for each condition and biological replicate. Graphs show mean values and standard deviations from the three biological replicates per condition. **Right:** Coverage plot illustrating the accumulated read mapping to the *Myotis daubentonii* contig NC_081844.1. The gray lines represent the read counts for each of the 18 samples. The blue line depicts the sum of reads across all samples, which is scaled by a factor of 0.1.

**Figure 3 viruses-16-01210-f003:**
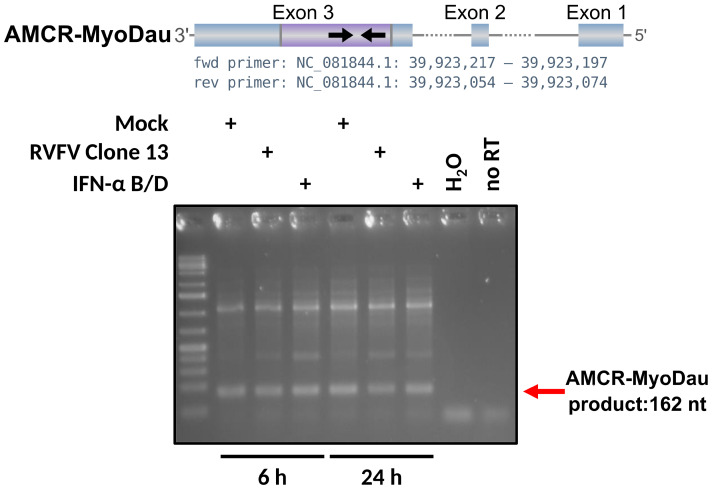
**Top:** Schematic figure of the three exons of AMCR-MyoDau, with EBLL-Cultervirus.10-MyoDau (CV.10-MyoDau) highlighted in purple, thus showing the primer pairs and positions used for RT-PCR analysis. **Bottom:** The expression of AMCR-MyoDau RNA in the *Myotis daubentonii* cell line is shown. The forward primer sequence is CTCCCTTGAGGAGTGTGACC, and the reverse primer sequence is GGCTGTCAGCAACAGTTTCA, thus targeting exon 3. Transcription of AMCR-MyoDau was observed under all available conditions, with no discernible impact on superinfection with RVFV Clone13 or IFN treatment. The *no RT* control consists of the Clone13 24 h RNA sample that underwent the RT step without adding primers.

**Figure 4 viruses-16-01210-f004:**
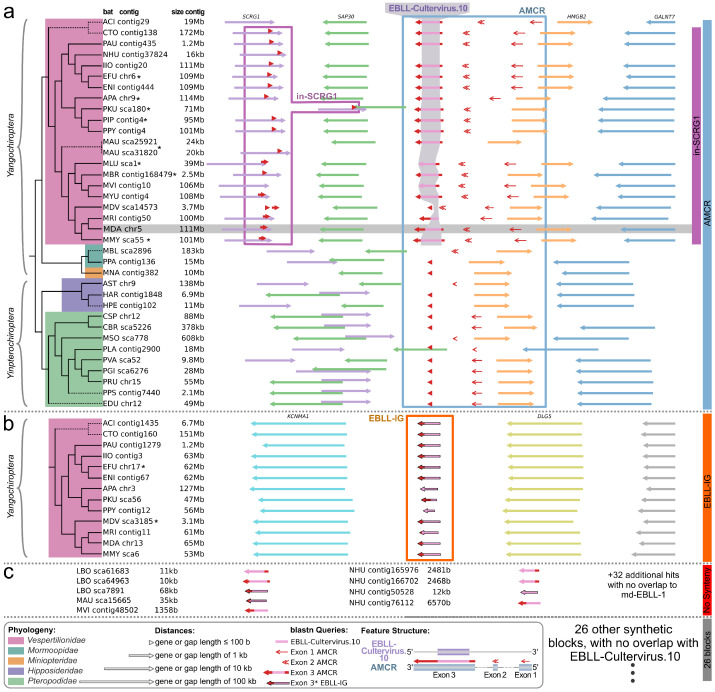
Syntenic analysis for AMCR ((**a**) in blue), EBLL-Cultervirus.10 ((**a**) in pink; short: CV.10), the previously described EBLL-IG (**b**), and homologs that could not be assigned to any syntenic pattern (**c**). Gene lengths, gap lengths, and the length of EVE featured are logarithmically scaled. The abbreviations for the bat species are explained in Table 1. The initial sequence queries are the three exons (three arrow types) of AMCR-MyoDau (gray box) and the sequence of the previously described EBLL-IG (NC_081852.1:16,169,651–16,170,474), which exhibited high similarities with exon 3 of AMCR-MyoDau. Pink bars in exon 3 indicate sequence similarity to CV.10, yet such similarities do not definitively signify orthology with CV.10. The syntenic genes of all homologs were analyzed and subsequently grouped into syntenic blocks (colored boxed on the right. The color-coded families of the taxonomic assignments (left) from Agnarsson et al. [60] are complemented with bat species absent in the Agnarsson study (dashed lines). A comprehensive list of all sequences and their synthetic associations is provided in the https://www.doi.org/10.17605/OSF.IO/89EF2 and visualized in Appendix A. Some orthologs, marked with an asterisk (*), were already identified using automated pipelines [48,72]. Bats marked with an asterisk (*) in the EBLL-IG clade have already been described in the literature [68].

**Figure 5 viruses-16-01210-f005:**
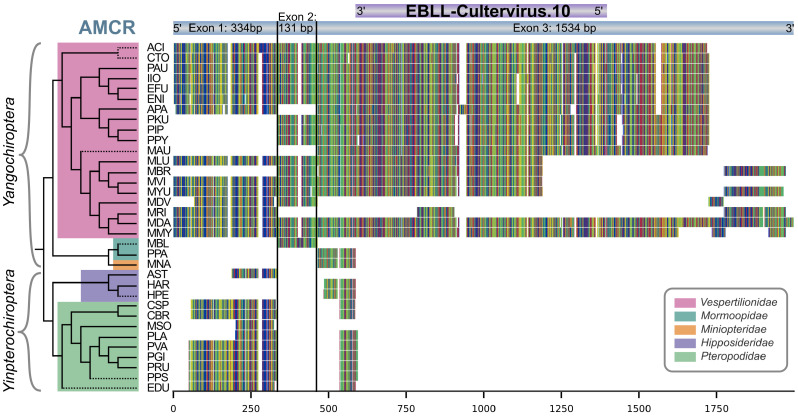
The schematic nucleotide alignment of AMCR is depicted from all sequences found in the bat genomes (see Table 1) whenever there was a commonality with the synteny to *Myotis daubentonii*. The exon structure and alignment lengths are depicted above the alignments. Taxonomic assignments derived from Agnarsson et al. are shown on the left margin [60]. Families are color-coded, and bat species not included in the study by Agnarsson et al. are represented with dashed lines in the tree. The abbreviations for the bat species are explained in Table 1.

**Table 1 viruses-16-01210-t001:** We selected 63 bat genomes of different assembly qualities. Genomes marked with an asterisk (*) are RefSeq genomes. We introduced a three-letter abbreviation for each bat species and listed, for each assembly, the total number of contigs (#con), how many of them represent complete chromosomes (#chr), the N50 value, and the total assembly size in gigabases (Gb) according to NCBI.

Species	Abb.	#Con|#Chr	N50	Size	NCBI Acc.	Year	Family
			[Mb]	[Gb]			
* **Yangochiroptera** *							
*Aeorestes cinereus*	ACI	2536|0	35.1	2.1	GCA_011751065.1	2020	*Vespertilionidae*
*Corynorhinus townsendii*	CTO	182|0	177.8	2.0	GCA_026230045.1	2022	*Vespertilionidae*
*Plecotus auritus*	PAU	5570|0	186.5	2.2	GCA_963455325.1	2023	*Vespertilionidae*
*Nycticeius humeralis*	NHU	1,676,240|0	0.015	2.8	GCA_007922795.1	2019	*Vespertilionidae*
*Ia io*	IIO	2008|0	105.8	2.1	GCA_025583905.1	2022	*Vespertilionidae*
*Eptesicus fuscus*	EFU	48|25	102.8	2.0	GCF_027574615.1 *	2023	*Vespertilionidae*
*Eptesicus nilssonii*	ENI	726|0	102.4	2.0	GCA_030846915.1	2023	*Vespertilionidae*
*Lasiurus borealis*	LBO	518,900|0	0.039	2.9	GCA_004026805.1	2019	*Vespertilionidae*
*Antrozous pallidus*	APA	93|23	114.6	2.1	GCA_027563665.1	2023	*Vespertilionidae*
*Pipistrellus kuhlii*	PKU	202|0	80.2	1.8	GCF_014108245.1 *	2020	*Vespertilionidae*
*Pipistrellus pipistrellus*	PIP	323|0	94.9	1.8	GCA_903992545.1	2020	*Vespertilionidae*
*Pipistrellus pygmaeus*	PPY	243|0	89.5	1.9	GCA_949987585.1	2023	*Vespertilionidae*
*Murina aurata feae*	MAU	880,177|0	0.026	2.3	GCA_004026665.1	2019	*Vespertilionidae*
*Myotis lucifugu*	MLU	11,654|0	4.3	2.0	GCF_000147115.1 *	2010	*Vespertilionidae*
*Myotis brandtii*	MBR	169,750|0	3.2	2.1	GCF_000412655.1 *	2013	*Vespertilionidae*
*Myotis vivesi*	MVI	64,503|0	91.8	2.1	GCA_035771395.1	2024	*Vespertilionidae*
*Myotis yumanensis*	MYU	476|0	99.1	2.0	GCA_028538775.1	2023	*Vespertilionidae*
*Myotis davidii*	MDV	101,769|0	3.5	2.1	GCF_000327345.1 *	2012	*Vespertilionidae*
*Myotis ricketti*	MRI	105|0	80	2.0	GCA_036010255.1	2024	*Vespertilionidae*
*Myotis daubentonii*	MDA	121|23	102.2	2.1	GCF_963259705.1 *	2023	*Vespertilionidae*
*Myotis myotis*	MMY	93|0	94.4	2.0	GCF_014108235.1 *	2020	*Vespertilionidae*
*Molossus nigricans*	MNI	146|0	81.9	2.4	GCA_026936385.1	2022	*Molossidae*
*Molossus alvarezi*	MAL	187|0	113.9	2.4	GCA_031001765.1	2023	*Molossidae*
*Molossus molossus*	MMO	60|0	110.7	2.3	GCF_014108415.1 *	2020	*Molossidae*
*Tadarida brasiliensis*	TBR	148|25	111.1	2.3	GCA_030848825.1	2023	*Molossidae*
*Rhynchonycteris naso*	RNA	50|0	286.1	2.4	GCA_031021685.1	2023	*Eallonurida*
*Sturnira hondurensis*	SHO	25,881|0	10.2	2.1	GCF_014824575.3 *	2022	*Phyllostomidae*
*Tonatia saurophila*	TSA	249,810	0.166	2.1	GCA_004024845.1	2019	*Phyllostomidae*
*Trachops cirrhosus*	TCI	396,519	124.5	2.2	GCA_028533065.1	2023	*Phyllostomidae*
*Micronycteris hirsuta*	MHI	550,090|0	0.069	2.3	GCA_004026765.1	2019	*Phyllostomidae*
*Carollia perspicillata*	CPE	1,925,339|0	0.010	2.7	GCA_004027735.1	2019	*Phyllostomidae*
*Anoura caudifer*	ACU	337,255|0	0.143	2.2	GCA_004027475.1	2019	*Phyllostomidae*
*Desmodus rotundus*	DRO	573|14	160.1	2.1	GCF_022682495.1 *	2022	*Phyllostomidae*
*Phyllostomus discolor*	PDI	78|17	171.7	2.1	GCF_004126475.2 *	2020	*Phyllostomidae*
*Phyllostomus hastatus*	PHA	534|0	39.2	2.1	GCF_019186645.2 *	2021	*Phyllostomidae*
*Macrotus californicus*	MCA	1,128,787|0	0.017	2.2	GCA_007922815.1	2019	*Phyllostomidae*
*Artibeus jamaicensis*	AJA	868|0	22.1	2.1	GCF_021234435.1 *	2021	*Phyllostomidae*
*Pteronotus parnellii*	PPA	333|0	31.5	2.1	GCF_021234165.1 *	2021	*Mormoopidae*
*Mormoops blainvillei*	MBL	205,259|0	0.156	2.1	GCA_004026545.1	2019	*Mormoopidae*
*Noctilio leporinus*	NLE	298,222|0	0.136	2.1	GCA_004026585.1	2019	*Noctilionidae*
*Miniopterus natalensis*	MNA	1269|0	4.3	1.8	GCF_001595765.1 *	2016	*Miniopteridae*
*Miniopterus schreibersii*	MSC	177,620|0	0.109	1.8	GCA_004026525.1	2019	*Miniopteridae*
* **Yinpterochiroptera** *							
*Megaderma lyra*	MLY	1,902,801|0	0.072	2.6	GCA_004026885.1	2019	*Megadermatidae*
*Craseonycteris thonglongyai*	CTH	1,224,256|0	0.026	2.3	GCA_004027555.1	2019	*Craseonycteridae*
*Aselliscus stoliczkanus*	AST	191|16	162	2.2	GCA_033961575.1	2023	*Hipposideridae*
*Hipposideros pendleburyi*	HPE	28,685|0	15.4	2.2	GCA_021464545.1	2022	*Hipposideridae*
*Hipposideros armiger*	HAR	7571|0	2.3	2.2	GCF_001890085.2 *	2016	*Hipposideridae*
*Hipposideros larvatus*	HLA	69|18	185.5	2.3	GCA_031876335.1	2023	*Hipposideridae*
*Rhinolophus ferrumequinum*	RFE	50|0	92	2.1	GCA_014108255.1	2020	*Rhinolophidae*
*Cynopterus sphinx*	CSP	181|17	145.2	1.9	GCA_030015415.1	2023	*Pteropodidae*
*Cynopterus brachyotis*	CBR	48,006|0	0.251	1.8	GCA_009793145.1	2019	*Pteropodidae*
*Macroglossus sobrinus*	MSO	171,985|0	0.453	1.9	GCA_004027375.1	2019	*Pteropodidae*
*Pteropus alecto*	PLA	65,598|0	6	2.0	GCF_000325575.1 *	2013	*Pteropodidae*
*Pteropus vampyrus*	PVA	36,094|0	6	2.2	GCF_000151845.1 *	2014	*Pteropodidae*
*Pteropus giganteus*	PGI	16,113|0	18.9	2.0	GCF_902729225.1 *	2020	*Pteropodidae*
*Pteropus rufus*	PRU	469,091|19	110.5	2.1	GCA_028533765.1	2023	*Pteropodidae*
*Pteropus pselaphon*	PPS	7513|0	0.770	1.9	GCA_014363405.1	2020	*Pteropodidae*
*Eidolon dupreanum*	EDU	1,191,098|17	101.6	2.3	GCA_028627145.1	2023	*Pteropodidae*
*Eidolon helvum*	EHE	133,538|0	0.028	1.8	GCA_000465285.1	2013	*Pteropodidae*
*Eonycteris spelaea*	ESP	4469|0	8	2.0	GCA_003508835.1	2018	*Pteropodidae*
*Rousettus madagascariensis*	RMA	1,467,186|18	85.8	2.3	GCA_028533395.1	2023	*Pteropodidae*
*Rousettus leschenaultii*	RLE	8141|0	32.7	1.9	GCA_015472975.1	2020	*Pteropodidae*
*Rousettus aegyptiacus*	RAE	29|0	113.8	1.9	GCF_014176215.1 *	2020	*Pteropodidae*

**Table 2 viruses-16-01210-t002:** For all EBLL-Cultervirus.10 elements (short: CV.10) confirmed by our synteny analysis, the table lists the EVE name, the species, the position, and whether the specific EVE has already been described in the literature.

Name EVE	Species	NCBI acc.	Start-Stop	Literature
CV.10-MyoDau	*Myotis daubentonii*	NC_081844.1	39,922,667–39,923,434	
CV.10-AeqCin	*Aeorestes cinereus*	JAAGEH010000014.1	12,648,154–12,648,930	-
CV.10-CorTow	*Corynorhinus townsendii*	JAPDVU010000006.1	98,426,944–98,427,716	-
CV.10-PleAur	*Plecotus auritus*	CAUOHH010000436.1	107,056–107,829	-
CV.10-IaIo	*Ia io*	JAJQQW010000006.1	73,804,507–73,805,270	-
CV.10-EptFus	*Eptesicus fuscus*	NC_072478.1	36,387,431–36,388,196	[48,72]
CV.10-EptNil	*Eptesicus nilssonii*	JAULJE010000005.1	36,472,464–36,473,231	-
CV.10-AntPal	*Antrozous pallidus*	CM050516.1	75,692,535–75,693,314	[72]
CV.10-PipKuh	*Pipistrellus kuhlii*	NW_023425584.1	64,293,707–64,294,472	[72]
CV.10-PipPip	*Pipistrellus pipistrellus*	LR862361.1	31,459,988–31,460,753	[72]
CV.10-PipPym	*Pipistrellus pygmaeus*	OX465307.1	64,495,766–64,496,530	-
CV.10-MurAur	*Murina aurata feae*	PVJC01025922.1	14,138–14,917	[72]
CV.10-MyoLuc	*Myotis lucifugus*	NW_005871049.1	34,243,866–34,244,427	[48,72]
CV.10-MyoBra	*Myotis brandtii*	NW_005370908.1	1,501,243–1,501,804	[48,72]
CV.10-MyoViv	*Myotis vivesi*	JAWPEG010000011.1	69,888,777–69,889,333	-
CV.10-MyoYum	*Myotis yumanensis*	JAPQVT010000005.1	70,588,059–70,588,617	-
CV.10-MyoRic	*Myotis ricketti*	JASKON010000005.1	71,087,272–71,087,392	-
CV.10-MyoMyo	*Myotis myotis*	NW_023416368.1	71,971,796–71,972,569	[72]

## Data Availability

The data are contained within the article or the Appendix A. The data presented in this study are available in the Appendix A through the Open Science Framework at https://www.doi.org/10.17605/OSF.IO/89EF2.

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
