# Peer review of "Endogenous Bornavirus-like Elements in Bats: Evolutionary Insights from the Conserved Riboviral L-Gene in Microbats and Its Antisense Transcription in Myotis daubentonii"

_viruses, 2024, doi:10.3390/v16081210_

Round 1

Reviewer 1 Report

Comments and Suggestions for Authors

In this manuscript, Ritsch et al. describe a newly identified EBL in microbats. The authors detected a new EBLL in the genome of M. daubentonii and analyzed it deeply. This EBLL is conserved among many microbat genomes, and a minus-stranded transcript is transcribed in at least cells derived from M. daubentonii. Although the biological function of this transcript is unknown, these findings could contribute to future EVE research and our understanding of bat immunity. Additionally, in the last paragraph, the authors raise a very important issue in the EVE research field. Thus, this paper would make an important contribution to the EVE research field. However, there are several problems with the writing and data presentation in the paper that make it difficult for the readers to understand (in fact, I found some parts difficult to understand). The authors need to address these issues. My concrete comments are as follows.

General comments

1.         In the Results and Discussion sections, there are many points where it is difficult for the readers to interpret and understand the results because of the inappropriate way the descriptions and data are presented. For example, in the Results section, there are many descriptions that do not explain what the authors have done and what the analysis was for. Actually, many of these are described in the "Materials and Methods" section, but it is not obvious for the readers which part they should read. As the authors wrote in L220, a brief description of the purpose and methods of each analysis would help the readers to understand them very easily. Please see below for specific comments on each figure.

2.         Although it may not be mandatory, the quality of this paper would be improved if the authors could indicate which regions are the authentic insertion. The BLAST hit regions do not represent the entire region of the authentic inserted EBL sequence. Fig. 5 shows that many Yinpterochiroptera bats appear to have mdEBLL-1-empty loci. Alignment with these genomic sequences may reveal more detailed information about the insertion. By identifying this, it may become clearer to what extent host genome-derived regions contribute to transcription/transcripts, and to what extent viral-derived sequences do so.

Specific comments

3.         L48-49: This description is inaccurate because, as far as I understand it, integrated virus-derived sequences inherited to the next generation are called EVE.

4.         L57-59: EBLX and EBLP have also been detected, although in fewer numbers. Please refer to the following papers (https://doi.org/10.1073/pnas.2026235118, https://doi.org/10.1093/ve/vead038).

5.         L72-74: The description is not accurate because syncytin genes are well described not only in humans but also in other mammals, especially mouse. Also, the original work should also be cited here.

6.         L100: This sequence is not from BDV but from Variegated squirrel bornavirus 1 (VSBV-1).

7.         L100: I am just curious why the authors chose VSBV-1 but not BoDV-1 as a query.

8.         L134: Which primer did the authors use for RT?

9.         L163, "EBLL-IG": this should be read "EBLL inter-genus (EBLL-IG)".

10.      L191-192: The authors should also check at least the following papers. Similar comprehensive EBLL detection has been done in these papers in more recent years. As far as I checked, mdEBLL-1 was not detected in these papers either, but the authors should check them themselves.
https://doi.org/10.1073/pnas.2026235118
https://doi.org/10.1101/2023.10.17.562709 (https://github.com/giffordlabcvr/DIGS-for-EVEs/blob/main/vertebrates/output/all-nonretroviral-eves-with-taxonomy.xlsx)

11.      L199: The authors should clearly mention which RNA-seq analysis they are describing here. Also, it would be more helpful to the readers if the authors could briefly describe the purpose and what they did with what data.

12.      L208- and Fig. 2: The authors wrote "shallow but constant expression," but it is not clear to what extent it is expressed. In Fig. 2, it would be better to indicate the expression level in tpm, which would be easier for the readers to understand.

13.      L220-224: Did the authors use a strand-specific primer for RT? If so, please clearly describe it.

14.      L220-224 and Fig. 3: I would recommend checking the sequence of bands of expected size.

15.      L225: The significance of this section (and Fig. 4) was unclear. As far as I understood, Fig. 4 is a genome-based analysis, not a transcript analysis. Therefore, it would be clearer to the readers if Fig. 4 is a supplementary figure to the analysis in Fig. 5. If I am missing some important intention of the authors, please explain it.

16.      L225: Since it is not possible to confirm the presence of transcripts from genome analysis, it would be better to change the title of the subsection to something appropriate.

17.      L225, Fig. 5, etc.: I was very confused when I read the manuscript because the authors named the antisense transcript as-mdEBLL-1. This confusion arises because the name EBLL is also used for regions not homologous to the viral L (with no BLAST hits). I would recommend giving a different name to sequences that consist only of regions that are not homologous to EBLL to prevent this confusion.

18.      L250, "we could also observe the already described EBLL-IG": This description is very vague. Please describe the details (As far as I understood, this is a BLAST hit, but is not an ortholog of mdEBLL-1).

19.      L254-257: It is unclear which data suggest a duplicate event.

20.      L261-265: Did the authors consider the possibility that exon3-like sequences are SINE?

21.      Discussion section: Please describe this section using the paragraph writing techniques.

22.      L297, "protein features": What does this mean?

23.      L300-301: Is there any data supporting this conclusion?

24.      L302: The authors did not show the exon structure is preserved among the bats. To conclude that the exon structure is conserved, transcripts in the other bats need to be analyzed.

25.      Fig. 1: "BDV" should be "VSBV-1". Also, please do not italicize the virus name "Variegated…", because this is not a species name.

26.      Fig. 2, left: Please show the expression levels as tpm.

27.      Fig. 2, right: I did not understand how to interpret the gray lines. How do I distinguish each sample?

28.      Fig. 3: Which sample did the authors use for "no RT"? This is not mandatory, but ideally we should do "no RT" on all the samples.

29.      Fig. 5: The blue box is not correct, because those of Yinpterochiroptera bats do not contain the EBLL region. This comment is related to Comment #19.

30.      Fig. 5: Some of "EBLL-IG"s are depicted in pink, but this is very confusing because it is not an authentic mdEBLL-1.

Reviewer 2 Report

Comments and Suggestions for Authors

The paper describes an investigation of a set of bornavirus-derived endogenous viral element (EVE) found in bat genomes. These EVEs, which are derived from the borna virus L-polymerase gene, are interesting for several reasons. Previous studies have shown they include some loci encoding relatively long open reading frames (ORFs) - i.e. 1700 codons - that have apparently been preserved by purifying selection for >12 million years, suggesting they have been functionalised in some way.

The present study reveals another interesting feature of these EVEs - that some encode an antisense transcript, the expression of which is conserved across multiple bat species. This is a surprising and interesting finding.

I am satisfied that the authors have taken a rigorous and appropriate approach to investigating the biology of these interesting EVEs. However, I do find there to be some issues with the manuscript which must be addressed before publication.

# Nomenclature

An important issue is the nomenclature applied here to bornavirus EVEs.

The authors state in their discussion that "one primary challenge in the still-young field of EVE research is the absence of a standardized database and nomenclature for their cataloging. Currently, there is no centralized repository documenting previously identified EVEs, complicating the process of verifying whether a newly discovered EVE is truly novel."

This is not entirely true. Standards for naming EVE loci have been proposed, and a centralised repository was recently published:

https://github.com/giffordlabcvr/DIGS-for-EVEs

In this repository, a standardised nomenclature convention developed for endogenous retroviruses (ERVs) has been applied to EVEs derived from non-retroviral viruses.

Probably the important aspect of this annotation is that it provides a way to distinguish orthologous and paralogous EVE loci, which is highly relevant to this study, where the authors refer to several different groups of endogenous bornavirus-like L-protein (EBLL) elements, but it is not entirely clear from the manuscript which ones are orthologous and which represent distinct germline incorporation events. Indeed, calling they element described here 'mdEBLL-1' makes it seem like an ortholog of a previously reported element 'ehEBLL-1', but if I understand correctly, it is not.

Most work on bornavirus EVEs has been performed by the Tomonaga lab, who also were the first to report the bat EBLL elements described here. The authors of the present report have basically extended the non-standardised naming convention applied in that previous report (Horie et al, 2016, Scientific Reports). However, in a more recent, comprehensive analysis of bornavirus EVEs (Kawasaki et al, 2021, PNAS), researchers working in the Tomonaga lab applied the standard nomenclature described above to all bornavirus EVEs identified in mammal genomes, including the bat EBLLs they had previously reported.

I appreciate that because there are many bornavirus EVEs in mammals, and syntenic relationships can be troublesome to resolve, there are likely to be shortcomings in the naming of EBLLs in previous studies (e.g. paralogous loci may have been mistaken for orthologs, or vice versa). Furthermore, there may be shortcomings of the proposed nomenclature. Nevertheless, in the interests of clarity and helping us move toward general improvements in the systematic naming of EVEs, I would fervently encourage the authors to try and match the EBLL loci they are reporting here to the standardised locus names reported in Kawasaki et al. Where/if novel EBLL loci have been identified, I would suggest to name them using standardised conventions.

# EBLL diversity

The authors fail to discuss the diversity of bornaviruses - for example, that there are multiple genera (orthoborna-, culter-, carbo-) and that EVEs derived from each of these genera occur. They provide no information about which of these bornavirus subgroups the elements they are examining derive from. While it may not be essential, one expects a study like this to include a phylogenetic tree that places the EVEs being discussed into evolutionary context with other EBLLs and with bornavirus L genes. Knowing which bornavirus subgroup the element derives from is also an essential step in naming it appropriately.

# Writing

Although the paper is generally well written, there are a few areas where words are missing or missed, and slightly inaccurate information is provided. I encourage the authors to revise the text carefully, making sure that they are saying exactly what they want to say. Below I have indicated some minor problems that need addressing.

* "Bats are ecologically diverse vertebrates, distinguished by unique characteristics such as a natural resistance to pathogenic viruses and the presence of numerous endogenous viral elements 2 (EVEs)*

I agree bats are diverse and have unique characteristics. However, the two things listed here are not good examples of these.

- Their 'natural resistance to pathogenic viruses' is more of a popular idea than a scientific fact.
- They are not the only vertebrate group to have numerous EVEs

* to learn vocal

- Missing word?

* "Different types of endogenous bornavirus-like (EBL) elements exist. To date, EBLN (nucleoprotein), EBLM (matrix protein), EBLG (envelope glycoprotein), and EBLL 8 (RNA-dependent RNA polymerase) are known"

- the authors could more correctly explain that these element 'types' are in fact derived from different bornavirus mRNAs, and briefly explain the integration mechanism.

* Yangochiroptera is a suborder not an an order

* Define GTF (Gene Transfer Format) file

* "Additionally, we found as-mdEBLL-1 significantly (adjusted 215 p-value < 0.05) differential expressed "

-differentially expressed?

* "This decline observed after 24 hours for Clone13 could be attributed to viral cap-snatching [68]."

Could the authors perhaps explain in a sentence what they mean?

* "In the depicted synteny plot of as-mdEBLL-1 (Figure 5), homologous of exon 3*

- homologs of exon 3?

* Once integrated into the host genome, EVEs can increase their population via vertical transmission

- increase their *allele frequency*?
- it is important to differentiate increases ingenue frequency EVE loci versus expansion of EVE copy number, which can also occur.

* "After extensive literature screening, to the best of our knowledge, we present a novel EVE, called"

- should probably be: After extensive literature screening we present a previously unreported EBLL locus, called".

* Figure 5 - I think the legend needs some work to make it clear

Comments on the Quality of English Language

See above

Round 2

Reviewer 1 Report

Comments and Suggestions for Authors

The manuscript has been significantly improved, but I still have some comments.

1. >8.  L134: Which primer did the authors use for RT?
>A: The primer sequences are mentioned in this section, see:  „PCR was conducted using a primer set designed using the online software Primer3Plus to detect the AMCR-MyoDau 90 nucleotide sequence of Exon 3 (fwd primer: CTCCCTTGAGGAGTGTGACC and rev primer: GGCTGTCAGCAACAGTTTCA), KOD polymerase (Calbiochem, 71086-3), and 2 µl of the respective cDNAs as templates.“

I asked about the primer for reverse-transcription but not PCR. But, the authors described that "A total of 100 ng isolated RNA was used for random primed cDNA synthesis using PrimeScript High Fidelity 146 RT-PCR Kit (Takara, R022B).", so they probably used random hexamer for reverse-transcription. If so, the PCR result does not support that the detected bands were derived from an antisense transcript because random hexamer could reverse transcribe both sense and antisense RNA. A primer specifically binding to the antisense transcript should be used for reverse transcription to support the authors' conclusion.

2. L212, "originating from the L protein of Wuhan sharpbelly bornavirus from the genus Culter-virus": this should be "originating from L gene of a cultervirus" or something like that because this EBL is not derived from protein and Wuhan sharpbelly bornavirus.

3. L304: The authors claim that they are paralogs due to gene duplication, but their data does not support this. It should be noted that even if sequence similarity is observed in the EBL regions, those EBLs may have formed by independent endogenization events of similar viruses.

Reviewer 2 Report

Comments and Suggestions for Authors

The authors have done a good job of addressing my concerns. I think it really helps the manuscript to use nomenclature consistent with Kawasaki et al. I am satisfied with the revised manuscript, but just wanted to mention to the authors - in case they are not aware - that it would acceptable to further abbreviating the element IDs to facilitate the flow of the text.

I note that - correctly - they have sometimes shortened the element ID by dropping the host component when talking generally about the locus, rather than a copy within a species:

i.e. so that EBLL-Cultervirus.10.MyoDau becomes just 'EBLL-Cultervirus.10'

I think it would also be perfectly acceptable - once the full element ID has been defined - to abbreviate within subcomponents:

e.g. so that 'EBLL-Cultervirus.10' becomes 'EBLL-CV.10' or 'EBLL-Culter.10'

Additionally, since EBLL elements are the only type of EVE being discussed in the paper, the authors could even consider dropping the EBLL component of the ID for the purposes of discussion.

e.g. EBLL-Cultervirus.10.MyoDau (hereafter referred to as 'CV.10.MyoDau')

But I leave this entirely up to the author's discretion.
